# Chimeric Stimuli-Responsive Liposomes as Nanocarriers for the Delivery of the Anti-Glioma Agent TRAM-34

**DOI:** 10.3390/ijms22126271

**Published:** 2021-06-10

**Authors:** Nikolaos Naziris, Natassa Pippa, Evangelia Sereti, Varvara Chrysostomou, Marta Kędzierska, Jakub Kajdanek, Maksim Ionov, Katarzyna Miłowska, Łucja Balcerzak, Stefano Garofalo, Cristina Limatola, Stergios Pispas, Konstantinos Dimas, Maria Bryszewska, Costas Demetzos

**Affiliations:** 1Section of Pharmaceutical Technology, Department of Pharmacy, School of Health Sciences, National and Kapodistrian University of Athens, Panepistimioupolis Zografou, 15771 Athens, Greece; niknaz@pharm.uoa.gr (N.N.); natpippa@pharm.uoa.gr (N.P.); 2Department of General Biophysics, Faculty of Biology and Environmental Protection, University of Lodz, Pomorska 141/143, 90-236 Lodz, Poland; marta.kedzierska@edu.uni.lodz.pl (M.K.); jakub.kajdanek@unilodz.eu (J.K.); maksim.ionov@biol.uni.lodz.pl (M.I.); katarzyna.milowska@biol.uni.lodz.pl (K.M.); 3Theoretical and Physical Chemistry Institute, National Hellenic Research Foundation, 48 Vassileos Constantinou Avenue, 11635 Athens, Greece; chrisostomou.varvara@gmail.com (V.C.); pispas@eie.gr (S.P.); 4Department of Pharmacology, Faculty of Medicine, University of Thessaly, 41500 Larissa, Greece; sereti_e@yahoo.gr (E.S.); ksdimas@yahoo.com (K.D.); 5Laboratory of Microscopic Imaging and Specialized Biological Techniques, Faculty of Biology and Environmental Protection, University of Lodz, Banacha 12/16, 90-237 Lodz, Poland; lucja.balcerzak@biol.uni.lodz.pl; 6Department of Physiology and Pharmacology, Laboratory Affiliated to Istituto Pasteur Italia, Sapienza University of Rome, Piazzale Aldo Moro 5, 00185 Rome, Italy; Stefano.garofalo@uniroma1.it (S.G.); cristina.limatola@uniroma1.it (C.L.); 7IRCCS Neuromed, Via Atinense 18, 86077 Pozzilli, Italy

**Keywords:** chimeric liposomes, functional, pH-responsive, TRAM-34, drug delivery, glioblastoma

## Abstract

Nanocarriers are delivery platforms of drugs, peptides, nucleic acids and other therapeutic molecules that are indicated for severe human diseases. Gliomas are the most frequent type of brain tumor, with glioblastoma being the most common and malignant type. The current state of glioma treatment requires innovative approaches that will lead to efficient and safe therapies. Advanced nanosystems and stimuli-responsive materials are available and well-studied technologies that may contribute to this effort. The present study deals with the development of functional chimeric nanocarriers composed of a phospholipid and a diblock copolymer, for the incorporation, delivery and pH-responsive release of the antiglioma agent TRAM-34 inside glioblastoma cells. Nanocarrier analysis included light scattering, protein incubation and electron microscopy, and fluorescence anisotropy and thermal analysis techniques were also applied. Biological assays were carried out in order to evaluate the nanocarrier nanotoxicity in vitro and in vivo, as well as to evaluate antiglioma activity. The nanosystems were able to successfully manifest functional properties under pH conditions, and their biocompatibility and cellular internalization were also evident. The chimeric nanoplatforms presented herein have shown promise for biomedical applications so far and should be further studied in terms of their ability to deliver TRAM-34 and other therapeutic molecules to glioblastoma cells.

## 1. Introduction

One of the main obstacles to solid tumor therapy is the poor pharmacokinetics of drug molecules [1]. Stimuli-responsive technology is a step further for drug delivery, assisting the release of anticancer agents in a specific spatiotemporal way. After a stimuli-responsive nanocarrier reaches to the tumor site, it responds to the extracellular and/or intracellular environments, releasing the incorporated bioactive substance in specific tissues or cell compartments, but also at a specific rate. This utility is designed based on the deviant physiological conditions that exist inside a tumor, compared with healthy tissues [2]. These nanocarriers are defined as “functional” and “smart” advanced drug delivery nanosystems (aDDnSs) and are very promising vehicles for co-delivering diagnostic and therapeutic agents [3,4].

Gliomas are the most frequent type of tumor in the brain tissue. Gliomas in adults usually belong to the so-called diffuse type, where the brain tissue is infiltrated by diffuse tumor cells, and have been classified as astrocytic, oligodendroglial and oroligoastrocytic tumors, depending on their phenotype. The World Health Organization (WHO) distinguishes gliomas based on their cell type and malignancy grade (I to IV) and glioblastomas are the most common and malignant type. Currently, standard glioma treatment utilizes surgical resection, followed by a combination of radiotherapy and a specific chemotherapeutic agent, temozolomide. Poor prognosis, frequent recurrences and short survival have led to a demand for new and innovative therapeutic approaches for this difficult and complex disease [5,6].

Compared with normal cells, glioblastoma cells exhibit different expression in certain ion channels that relate to hallmarks of their aggressiveness. These channels mostly regulate the cell volume and intracellular Ca^2+^ concentration. Data on the intermediate-conductance calcium-activated potassium channel KCa3.1 (one of the three KCa channels) and on its altered expression and function in glioblastoma cells have recently become available and are very important for developing new therapeutic strategies [7,8,9]. The channel is inhibited by clotrimazole and 1-[(2-chlorophenyl) diphenylmethyl]-1H-pyrazole (TRAM-34). Glioblastoma treatment with TRAM-34 is not well-established yet and this is evident in the limited number of studies in the literature. However, this molecule is very promising in this regard, since its significant role in glioblastoma cell migration and consequent infiltrative behavior has been demonstrated both in vitro and in vivo [10,11,12,13].

Despite the promising therapeutic profile, there is no appropriate vehicle for TRAM-34, which leads to the utilization of lipophilic carriers. On top of that, the administration of very high doses in vivo is discouraging. In addition, the molecule has been associated with cytochrome P450 (CYP) activity inhibition in high doses, limiting its potential for co-therapy with other drugs [14]. A study has demonstrated that the binding site of the molecule resides in the cytoplasmic side of the cell membrane and interaction occurs only when TRAM-34 is applied from the inside [8,15,16]. Hence, its delivery inside the cell is of critical importance and can be assisted by functional nanocarriers. Specifically, the therapeutic profile of TRAM-34 may be improved by assisting its targeting to the glioblastoma site and facilitating its permeation through the cell membrane to the intracellular environment, where it can bind to the KCa3.1 ion channel. This might be achieved, for example, through pH-responsive nanocarriers, which will be activated in the low pH of early or late endosomes and promote the drug release inside the cell [17]. In this way, the dosage and observed adverse effects of the molecule may be reduced.

The aim of the present study was to design, develop and biologically evaluate pH-responsive chimeric aDDnSs composed of the phospholipid L-α-phosphatidylcholine (Egg, Chicken) (EPC) and pH-responsive amphiphilic diblock copolymers poly(2-(dimethylamino)ethyl methacrylate)-b-poly(lauryl methacrylate) (PDMAEMA-b-PLMA) with the incorporated drug molecule TRAM-34 (Figure 1). EPC is established in the literature, as well as in the market of liposomal medicines [18,19]. Its gel-to-liquid crystalline phase transition (*T_m_*) has been identified around −5 °C [20]. PDMAEMA-b-PLMA copolymers have been previously utilized for the development of chimeric nanosystems with added value in their physicochemical characteristics (i.e., colloidal stability, etc.) [21,22]. PDMAEMA is a well-established polyelectrolyte that has been extensively studied for the formation of nanocomplexes with DNA and gene delivery, and exhibits pH-responsive properties, due to the pK_a_ of amino groups being around 7.5–8.0 [23,24,25]. PLMA is a hydrophobic polymer, expected to serve as an anchor for the attachment of the copolymer onto the lipid bilayer. To the best of our knowledge, this is the first attempt to incorporate TRAM-34 into nanocarriers and deliver it inside glioma cells.

## 2. Results and Discussion

### 2.1. Membrane Fluidity of Chimeric Bilayers

The assessment of the effect of the PDMAEM-b-PLMA copolymers on EPC membrane fluidity was achieved using steady-state fluorescence anisotropy (FA) (Figure 2). Two types of probes were utilized. 1,6-diphenyl-1,3,5-hexatriene (DPH) is a non-polar molecule that is incorporated inside the hydrophobic region of the liposome bilayer with its long axis parallel to the acyl chains, whereas 1-[4-(trimethyl-ammonium) phenyl]-6-phenyl-1,3,5-hexatriene (TMA-DPH) is anchored onto the bilayer, with its positively charged amino groups in contact with the medium water molecules [26]. These two molecules allow for evaluation of the mobility of both the hydrophobic and the hydrophilic regions of the membrane, thus offering a complete picture of the membrane order at different depths of the bilayer [26].

The hydrophobic part of EPC bilayers was affected in terms of membrane fluidity by the progressive addition of either copolymer, reflected in the DPH anisotropy (Figure 2A). In every case, anisotropy was higher than the EPC reference, which means that both polymers are incorporated inside the membrane and interact with it at all concentrations. Specifically, up to a 0.5 polymer molar ratio the interaction becomes gradually higher, resulting in increasing DPH anisotropy and decreasing membrane fluidity. In addition, PDMAEMA-b-PLMA 1 (PDMAEMA1) seems generally to affect the membrane fluidity more than PDMAEMA-b-PLMA 2 (PDMAEMA2). However, at a molar ratio of 9:1, the anisotropy due to PDMAEMA-b-PLMA 1 is reduced, which indicates three possible scenarios regarding the copolymer: (i) its amount results in fluidity that is close to EPC, with increased cooperativity between the phospholipids and the copolymer molecules; (ii) it self-assembles in polymeric structures that do not reach the membrane interior; (iii) it creates polymer-rich domains that generate more space between the phospholipid chains and allow for more intense spin of DPH [21,26].

The polar region of the EPC bilayers was affected in a different way by the two polymers, compared with the hydrophobic region (Figure 2B). Low concentrations of the copolymers, up to 0.1 for PDMAEMA-b-PLMA 1 and 0.05 for PDMAEMA-b-PLMA 2, brought about a decrease in the anisotropy of TMA-DPH, indicating lipid–polymer interactions that led to increased mobility of the polar head groups or domain formation. At higher concentrations, they resulted in increased anisotropy/decreased head group mobility, with PDMAEMA-b-PLMA 2 reversing its effect at a 9:1 molar ratio and going below the anisotropy of EPC. It is important to note that PDMAEMA-b-PLMA 1 never exceeds the EPC anisotropy value, but by increasing the concentration, the mobility decreases gradually to reach EPC again. As a result, this molecule always results in the same or greater head group fluidity than EPC. On the other hand, PDMAEMA-b-PLMA 2 reduces the mobility at 9:0.2 and even more at 9:0.5, not allowing for probe spin at those concentrations. At 9:1, it either improves the fluidity again or it forms polymeric assemblies that do not reach the membrane–aqueous medium interface, not affecting the head groups at all [21,26].

### 2.2. Thermotropic Behavior of Chimeric Bilayers with TRAM-34

An essential aspect of liposomal development and especially chimeric nanosystems is the thermodynamics of membranes and the way their thermotropic behavior and functionality is altered by incorporating foreign biomaterials, such as drug molecules and amphiphilic copolymers [27,28]. The interactions between 1,2-dipalmitoyl-sn-glycero-3-phosphocholine (DPPC), TRAM-34 and PDMAEMA-b-PLMA 1 and 2 in physiological (phosphate buffer saline, PBS) and acidic environment (citrate buffer) were studied by means of differential scanning calorimetry (DSC) analysis and the diagrams are presented in Figure 3 for heating and Appendix A for cooling, whereas the calorimetric parameter values are listed collectively in Appendix A. Since EPC is a phospholipid mixture and does not exhibit a distinct transition curve, we chose DPPC for this study, which has a *T_m_* around 41 °C [29].

First, the incorporation of the drug molecule inside the membrane led to interactions with the polar heads and the elimination of the pre-transition of DPPC membrane phospholipids in PBS (Figure 3A) [30]. Chimeric bilayers were nevertheless expected to have no pre-transition, because of the polymer’s effect on the membrane [22]. Concerning the main transition, we observed that the incorporation of TRAM-34 led to a slight decrease in the *T_m_*, an increase in the Δ*T_1_*_/*2*,*m*_ and a slight increase in the Δ*H_m_* for all systems, indicating the thermotropic effect of the drug molecule on the membranes. In the case of DPPC, the main transition curve was influenced the least by the drug molecule, potentiating the absence of the drug molecule from the membrane interior or its distribution in a more homogeneous manner. However, in the case of the chimeric systems, the formation of a shoulder on the main transition, along with all other alterations, is an indication of the creation of drug-related domains [31].

After exposure to an acidic environment, the drug-loaded chimeric bilayer transitions exhibited no shoulders and the transition enthalpy Δ*H_m_* decreased, particularly for DPPC:PDMAEMA-b-PLMA 1 (Figure 3B). In addition, the membrane cooperativity improved for the first polymer, but decreased for the second, as indicated by the Δ*T_1_*_/*2*,*m*_ values. According to our previous study, it would be generally expected to observe lower cooperativity in bilayers after acidic exposure, due to pH-responsive membrane disruption; however, the disappearance of the drug-related shoulder led to the opposite result for the first polymer [22]. These observations combined, we assume that TRAM-34 is released from the chimeric membranes in acidic conditions, based on the divergent behavior between drug-loaded and reference systems.

### 2.3. Physicochemical Characteristics, Stimuli-Responsiveness and Protein Interactions of Chimeric Nanocarriers

The physicochemical properties of EPC, EPC:PDMAEMA-b-PLMA 1 and 2, with or without incorporated TRAM-34, are presented in Table 1. The size distributions of conventional and chimeric nanocarriers are presented in Appendix A. For each property, i.e., size, polydispersity and zeta potential, the variation under different temperatures and pH conditions was evaluated for the nanocarriers (Figure 4). For conventional liposomes, the hydrodynamic diameter was calculated at around 150 nm, with the polydispersity index (PDI) being 0.370. When loaded with TRAM-34, EPC liposomes increased in size by 15 nm and polydispersity slightly decreased. Notably, after a while, the suspension led to precipitation, obviously because of instability of the system and consequent aggregation.

After the insertion of the amphiphilic diblock copolymers in the EPC system, the nanoparticle size was retained at 150 nm. The polydispersity was greatly reduced, to 0.160 for EPC:PDMAEMA-b-PLMA 1 and to 0.220 for 2 [32]. At the same time, the ζ-potential increased to about 12.0 and 8.0 mV for the two copolymers, respectively. This is attributed to the positive charge that the PDMAEMA amino groups carry in PBS, due to their pK_a_ being approximately 7.5–8.0 [24,25]. TRAM-34-loaded chimeric liposomes were stable in due time in terms of particle size, whereas precipitation did not occur for up to 15 days (data not shown).

The effect of increasing temperature on liposomal size is presented in Figure 4A. We observe that by increasing temperature from 25 °C to 37 °C, the size underwent an increase as well. This behavior has been attributed to the energy content of liposomal membranes at high temperatures and the consequent vesicular volume increase [33]. Concerning acidic conditions, a slight increase in the chimeric nanosystems’ size in pH = 4.5 was observed, presumably owing to the PDMAEMA chains stretching after protonation of the amino groups, whereas EPC liposomes remained the same (Figure 4B). The most profound effect of a low pH environment on these systems was identified in the ζ-potential values. Specifically, their charge doubled, which is attributed to the increased protonation of the PDMAEMA amino groups in these acidic conditions [24,25].

The developed chimeric liposomes were also tested for their binding with serum proteins, through a protocol involving incubation with fetal bovine serum (FBS) medium. The results are presented in Figure 4C. Positively charged nanocarriers are substrates for extensive protein adsorption, such as bovine serum albumin (BSA) [34,35]. After incubation, EPC liposomes remained almost stable, with only a slight size increase, whereas chimeric ones presented a size increase of 15–20 nm. This outcome was more or less expected, since EPC membranes are neutral in terms of charge, whereas EPC:PDMAEMA-b-PLMA liposomes possess a positive surface charge and absorb more proteins, which increase their apparent size. Polydispersity also increased, with PDI rising by less than 0.100 for EPC and between 0.200–0.250 for chimeric nanoparticles. Finally, the zeta potential remained unaltered for EPC, but decreased by 15–20 mV for chimeric membranes, indicating the creation of a protein corona that surrounds the vesicles after protein adsorption [36]. The difference between the two utilized copolymers is attributed to the double *M_w_* of the PDMAEMA-b-PLMA 2 copolymer, associated with a greater number of positively charged amino groups.

Transmission electron microscopy (TEM) images of empty and TRAM-34-loaded nanocarriers are presented in Figure 5. In these images, we were able to assess the structure and morphology of the nanosystems, though several membranes probably collapsed due to the drying process [37,38]. EPC liposomes loaded with TRAM-34 could not be evaluated, since they were unstable after preparation. EPC liposomes are sphere-like objects with sizes greatly varying below and above the 200-nm scale bar value (Figure 5A). The membranes, lamellae and inner hydrophilic cores were more easily observed in the cases of chimeric liposomes (Figure 5B,C). Particles were unilamellar, small (below 200 nm) and more uniform in size, compared with EPC [39,40]. Concerning chimeric nanoparticles loaded with TRAM-34, we were unable to obtain good quality images, particularly for EPC:PDMAEMA-b-PLMA 2, possibly due to the nature of the drug. For EPC:PDMAEMA-b-PLMA 1, however, we could observe vesicular morphologies, and many membrane fragments or worm-like micelles were also visible (Figure 5D). The latter structures and their origins in systems that contain lipid and polymer biomaterials in certain molar ratios have been thoroughly discussed in a previous study [21].

### 2.4. Drug Entrapment Efficiency % (EE%) and Release in Acidic Conditions

The initially utilized amount of TRAM-34 in the chimeric nanosystems was designed to result in a concentration of 1.2 mg/mL in PBS, serving for in vitro studies, and 20 mg/mL of total biomaterials, i.e., phospholipid and copolymer, were in each case used to formulate that amount. However, each chimeric nanosystem behaved differently. In particular, with EPC:PDMAEMA-b-PLMA 1, we achieved an IE% of 73%, whereas with EPC:PDMAEMA-b-PLMA 2, it was 62% (Table 2). The difference in the architecture between the two copolymers was determinant for the final drug incorporation, with PDMAEMA-b-PLMA 1 being more hydrophilic and shorter as a molecule, probably allowing for higher amounts of the drug to reside inside the lipid bilayer, which was also evident through DSC analysis [41,42]. All biological assays were carried out by utilizing the initial formulations.

The drug molecule release profiles for the two chimeric nanosystems in physiological and acidic conditions are presented in Figure 6. Based on the data, we drew conclusions on the behavior of these nanocarriers, which is defined by the pH-responsive behavior of PDMAEMA chains in acidic environments. First, there is a difference in the release profiles between the two chimeric nanocarriers in PBS, whereas both exhibited burst release phenomena in the first hour. EPC:PDMAEMA-b-PLMA 1 released 40% of the drug after 1 h and then slowly reached up to 55% after 6 h (Figure 6A), whereas EPC:PDMAEMA-b-PLMA 2 released around 30% after 1 h and a total of 37% after 5 h (Figure 6B). This behavior is probably due to the different architecture of the utilized block copolymer, with the first being smaller and relatively more hydrophilic, whereas the second was double in *M_w_* and more hydrophobic. These properties create different phases inside the chimeric bilayer and lead to different interactions of the copolymer hydrophobic block with TRAM-34, as indicated by DSC analysis. The cumulative release did not increase after 24 h for both nanocarriers.

Concerning acidic conditions, the picture was almost the same for both nanosystems in each pH studied. A drug burst release of around 70% was achieved after the first 15 min, followed by 80% for EPC:PDMAEMA-b-PLMA 1 and 85% for EPC:PDMAEMA-b-PLMA 2 after 30 min and ending up at almost 100% after 1 h for both nanosystems. The difference between physiological conditions and acidic ones is due to the pH-responsive properties of the prepared nanosystems, where the PDMAEMA groups around the drug-loaded membrane become more protonated below their pK_a_ value (~7.5–8.0), mobilize the PLMA segments and perturb the membrane, leading to the rapid release of a high amount of the drug [24,25,43,44]. The drug release degree and kinetic profile is the same for all pH values, because PDMAEMA is almost completely protonated at 6.5 and below, based on its pK_a_.

### 2.5. In Vitro Toxicity and Uptake of Chimeric Nanocarriers by HEK-293 Cells

The toxicity of nanoparticles, also called “nanotoxicity”, is an issue that concerns researchers greatly, because it is the main obstacle in the further development of nanomedicines. Certain methods have been widely utilized to evaluate nanocarrier in vitro toxicity, addressing cell toxicity, immunotoxicity and genotoxicity, such as the 3-(4,5-dimethylthiazol-2-yl)-2,5-diphenyltetrazolium bromide (MTT) and lactate dehydrogenase (LDH) assays. In addition, different types of normal and cancer cells have served this purpose [45]. However, there are some limitations in this approach, which emanate from the incompatibility between the nature of the nanoparticles, e.g., liposomes, and the chemical basis of these methods [46]. Herein, we assessed the nanocarrier and TRAM-34-loaded nanocarrier toxicity via the AlamarBlue^®^ assay. The % cell viability of HEK-293 cells after their exposure to increasing concentrations of the nanosystems is presented in Figure 7.

At first glance, all nanocarriers, including EPC liposomes and EPC:PDMAEMA-b-PLMA chimeric liposomes, loaded with TRAM-34 or not, are non-toxic up to 30 μg/mL total biomaterial concentration. The concentrations of the drug molecule were 0.3, 0.8, 3.3, 5.0, 16.7 and 33.3 μM for the nanocarrier concentrations 2, 5, 20, 30, 100 and 200 μg/mL, respectively. Apparently, neither the nanocarrier nor the incorporated drug induced toxic effects on HEK-293 cells up to 30 μg/mL carrier concentration. Above that, EPC liposomes maintained their lack of toxicity. On the other hand, chimeric liposomes, with or without TRAM-34, appeared to induce cell toxicity at concentrations of 100 and 200 μg/mL, especially EPC:PDMAEMA-b-PLMA 1. The concentrations of the polymers in these cases were around 33 and 66 μg/mL for the first and 50 and 100 μg/mL for the second, respectively. We assume that the toxicity can possibly be attributed to the high amount of polymer in the microenvironment of the cells, which is cationic and interacts with the cellular membrane, especially for the first, which has more cationic groups, as well as to the cellular uptake of these cationic nanoparticles (see below) [47]. The toxicity of PDMAEMA on HEK-293 cells has been investigated before, where the *M_w_* of the polymer was much higher than herein (354,000 g/mol) and the polymer alone exhibited a much higher toxicity, at 15 and 20 μg/mL after 24 h [48].

The cellular uptake of the chimeric liposomes, labeled with rhodamine B, by HEK-293 cells after 24 h incubation was evaluated through confocal laser scanning microscopy (CLSM) and is presented in Figure 8. For this assessment, we utilized the nanocarriers in a total biomaterial concentration of 30 μg/mL, corresponding to 1.72 μg/mL (5 μM) of TRAM-34, which was the tested concentration for antiproliferative effects on glioma cells in previous studies [11]. The confocal microscopy images show that the cells internalize the chimeric liposomes. This is achieved probably through electrostatic interaction between the positively charged nanocarriers and the negatively charged cell membrane, leading to adsorptive endocytosis [49,50]. In addition, the nanoparticles did not seem to end up inside the nucleus to a high degree. Instead, in some cases they were observed close to the inner side of the cell membrane (Figure 8B) [8,51,52].

### 2.6. In Vitro Antiproliferative Effect and Uptake of Drug-Loaded Chimeric Nanocarriers by GL261 Cells

The chimeric nanocarriers with incorporated TRAM-34 were evaluated for their effect on the GL261 murine glioma cell line, through the MTT assay. Glioma cells overexpress the intermediate-conductance KCa3.1 channel and its blockade by the drug/inhibitor increases cell apoptosis, whereas other effects include a decrease in colony formation, etc. [53]. The viability results of GL261 cells after 24 h, 48 h and 72 h incubation with drug, empty and drug-loaded nanocarriers are provided in Figure 9. The TRAM-34 concentration was 1.72 μg/mL (5 μM), whereas that of the nanocarriers was 30 μg/mL in all cases [11].

First of all, the effect of the drug molecule TRAM-34 on the GL261 proliferation was consistent for all three timepoints. This means that it always leads to around 70–75% cell growth at 5 μΜ, compared with the control, regardless of the duration of exposure. This antiproliferative effect on GL261 cells has been previously reported [11]. Serving as controls for the treatment, empty chimeric nanocarriers of a biomaterial concentration of 30 μg/mL were also tested for their effect on cell growth and were found to have no effect after 24 h, a slight effect after 48 h and a high effect after 72 h, leading to almost 30% growth, compared to the control. This time-dependent profile must be related with the cellular uptake of these nanocarriers by the particular cells, which is high, even after 24 h (see below). On this basis, EPC:PDMAEMA-b-PLMA 1:TRAM-34 was more toxic than neat TRAM-34 after 24 h, leading to 60% growth, whereas EPC:PDMAEMA-b-PLMA 2:TRAM-34 was less efficient, leading to 80% growth. After 48 h, the effect of both drug-loaded nanocarriers was the same with 24 h, whereas the effect of the empty carriers was perceptible. Finally, after 72 h, the loaded nanocarriers allowed for 20% growth, whereas the empty nanocarriers led to 30% growth, compared to the control. Conclusively, the antiproliferative effect of the EPC:PDMAEMA-b-PLMA 1:TRAM-34 chimeric nanosystem on glioma cells is slightly higher than the control and the second drug-loaded nanocarrier, whereas the nanocarriers are toxic for GL261 cells after long-term exposure.

The cellular uptake of chimeric liposomes, labeled with rhodamine B, by GL261 murine glioma cells after 24 h incubation was investigated through fluorescence microscopy (FM). The images are provided in Figure 10, together with the control. We observed in the images that both chimeric nanocarriers were localized inside the glioma cells. The interesting aspect of their behavior is that although they are internalized by these cells, probably through adsorptive endocytosis, after 24 h, they do not exert any toxic effects right away. However, it takes 48 h and even 72 h to produce their toxic effects (Figure 9).

### 2.7. In Vivo Toxicity of Chimeric Nanocarriers

EPC:PDMAEMA-b-PLMA 1 chimeric liposomes, which were more effective in delivering TRAM-34 inside GL261 cells, were evaluated for their acute toxicity in immunocompromised male NOD/SCID mice [54]. The formulation of 20 mg/mL was administered intraperitoneally (i.p.) in a single injection to mice at 100, 200, 300, 400 and 500 mg/kg and their behavior was monitored and recorded for approximately 5 h. Subsequently, they were weighed and observed for any sings of toxicity or routine alterations for a 15-day period.

The mice did not show any signs of sedation or abnormal behavior post-injection or during the next days in which they were monitored. In addition, doses up to 300 mg/kg led to no significant weight deviations during the study. After 10 days, one of the mice injected with 300 mg/kg presented an ulcer at the injection site, which healed a few days later. Concerning 400 and 500 mg/kg doses, those led to significant (over 10%) weight losses, with the mice returning to normal after a few days, whereas one of the 400 mg/kg doses was lethal. Skin ulcers were observed in these cases after 10 and 5 days for 400 and 500 mg/kg respectively, which also healed in the following days.

## 3. Materials and Methods

### 3.1. Materials

The saturated phospholipid DPPC, the phospholipid EPC and 1,2-dioleoyl-sn-glycero-3-phosphoethanolamine-N-(lissamine rhodamine B sulfonyl) (ammonium salt) (Rhod-PE) were purchased from Avanti Polar Lipids Inc. (Alabaster, AL, USA) and used without further purification. Chloroform and other reagents were of analytical grade and purchased from Sigma-Aldrich Chemical Co (St. Louis, MO, USA). The diblock copolymer PDMAEMA-b-PLMA was synthesized in two different molar compositions, 70–30% for PDMAEMA-b-PLMA 1 and 60–40% for PDMAEMA-b-PLMA 2, through reversible addition-fragmentation chain-transfer (RAFT) polymerization. The synthesis has been previously described [22]. The *M_w_* of the copolymers was equal to 6497 and 14,143 respectively. The drug molecule TRAM-34 had an *M_w_* of 344.84 and its synthesis has been previously described [16].

### 3.2. Preparation of Chimeric Bilayers

Pure lipid and chimeric lipid-block copolymer bilayers with or without the drug molecule TRAM-34 were prepared by mixing the appropriate amounts of EPC or DPPC, PDMAEMA-b-PLMA 1 or 2 and TRAM-34 in CHCl_3_:MeOH 9:1 *v*/*v* solutions and subsequently evaporating the solvent under vacuum and heat conditions, using a rotary evaporator (Rotavapor R-114, Buchi, Flawil, Switzerland). The molar ratio was between 9:0 and 9:1 for EPC:PDMAEMA-b-PLMA 1 or 2, 9:1.2 for DPPC:TRAM-34, 9:0.5 for DPPC:PDMAEMA-b-PLMA 1 or 2, 9:0.5:1.75 for DPPC:PDMAEMA-b-PLMA 1:TRAM-34 and 9:0.5:2.45 for DPPC:PDMAEMA-b-PLMA 2:TRAM-34. The polymer components were calculated based on the phospholipid and the drug was calculated in all cases to result in the same mass ratio between the drug and the total nanocarrier biomaterial, i.e., 6%. The vacuum applied was −1 atm and temperature was 40 °C. The films were maintained under these conditions until total evaporation and an additional 30 min and were then placed inside a desiccator, for at least 24 h, in order to remove any remaining traces of solvent. The obtained laminated bilayers were then hydrated and studied by means of FA and DSC.

### 3.3. Fluorescence Anisotropy/Polarization (FA)

The fluorescence anisotropy of two fluorescent probes, DPH and TMA-DPH, interacting with EPC:PDMAEMA-b-PLMA 1 or 2 chimeric bilayers of molar ratios between 9:0 and 9:1, was measured by using a PerkinElmer LS-50B spectrofluorometer (Waltham, MA, USA). For these measurements, the prepared bilayers (see Appendix A) were hydrated with PBS 0.15 M (pH = 7.4) to a final lipid concentration of 50 μM. Afterwards, DPH or TMA-DPH was added at a concentration of 1 μM; the suspension was vortexed and left to anneal for 5 min. For the measurements, the excitation wavelengths were 348 nm and 358 nm and the emission wavelengths were 426 nm and 428 nm for DPH and TMA-DPH, respectively. The slit width of the excitation monochromator was 2.5 nm and that of the emission monochromator was 5.5 nm for DPH and 20 nm for TMA. The fluorescence anisotropy values were calculated from Jablonski’s equation:(1)$$r=\frac{I_{VV}-GI_{VH}}{I_{VV}+G_{VH}}\;\;\;\;$$
where *r* = fluorescence anisotropy, and *I_VV_* and *I_VH_* = the vertical and horizontal fluorescence intensities, respectively, to the vertical polarization of the excitation light beam used. *G* = *I_VH_*/*I_VV_* (grating correction factor) corrects the polarization effects of the monochromator. The measurements were performed with Perkin Elmer software.

### 3.4. Differential Scanning Calorimetry (DSC)

DSC experiments were performed with a DSC822^e^ (Mettler-Toledo, Schwerzenbach, Switzerland) calorimeter, calibrated with pure indium (*T_m_* = 156.6 °C). Sealed aluminum crucibles of 40 μL capacity were used as sample holders. The prepared chimeric bilayers of DPPC, PDMAEMA-b-PLMA 1 or 2 and TRAM-34 were analyzed by placing approximately 3.0 mg of each sample in a crucible, hydrating with 20 μL of PBS 0.15 M (pH = 7.4) or citrate buffer 0.1 M (pH = 4.5), sealing and leaving samples to anneal for 30 min. Two heating-cooling cycles and a third heating scan were performed, to ensure good reproducibility of the data, with an empty aluminum crucible as reference. The temperature range was between 20 °C and 60 °C, whereas the scanning rate was 5 °C/min. Before each cycle, the samples were subjected to a constant temperature of 20 °C, to ensure equilibration. The calorimetric data obtained (characteristic transition temperatures *T_onset_*_,*m*/*s*_ and *T_m_*_/*s*_, enthalpy changes Δ*H_m_*_/*s*_ and widths at half peak height of the *C_p_* profiles Δ*T_1_*_/*2*,*m*/*s*_) were analyzed with Mettler-Toledo STAR^e^ software. All transition enthalpies were normalized per total biomaterial mass, including phospholipid and polymer, and were expressed as negative values for endothermic processes (during heating) and as positive values for exothermic ones (during cooling).

### 3.5. Preparation of Chimeric Nanocarriers

Chimeric liposomes of EPC, PDMAEMA-b-PLMA 1 or 2 and TRAM-34 were developed by utilizing the thin-film hydration method. Specifically, appropriate amounts of phospholipid, copolymer and drug molecule were dissolved in CHCl_3_:MeOH 9:1 and then transferred into a round-bottom flask. The molar ratio between the biomaterials was 9:1.2 for EPC:TRAM-34, 9:0.5 for EPC:PDMAEMA-b-PLMA 1 or 2, 9:0.5:1.75 for EPC:PDMAEMA-b-PLMA 1:TRAM-34 and 9:0.5:2.45 for EPC:PDMAEMA-b-PLMA 2:TRAM-34. The drug was calculated in all cases to result in the same drug–nanocarrier mass ratio, i.e., 6% *w*/*w*. The flask was connected to a Laborota 4000 rotary evaporator (Heidolph, Schwabach, Germany), a vacuum of −1 atm was applied and the thin films were formed by slow removal of the solvent at 40 °C. Then, they were maintained under these conditions for 30 min and finally, under vacuum for at least 24 h in a desiccator, to remove traces of solvent. Afterward, they were hydrated with PBS 0.15 M (pH = 7.4), by slowly stirring and heating for 1 h in a water bath, at 35 °C, above the phase transition temperature (*T_m_*) of EPC. The final biomaterial concentration of the chimeric systems was 20 mg/mL in each case, in respect to the lipid and polymer, whereas the drug concentration was 1.2 mg/mL or 3.48 mM. The resultant suspensions were subjected to extrusion through polycarbonate membranes, utilizing an AE-10 liposome extruder (ATS Engineering Limited, Suzhou, China) to obtain small unilamellar vesicles (SUVs). Specifically, the suspensions were subjected to 5 cycles of extrusion through a 400-nm pore membrane, followed by 5 cycles of extrusion through a 200-nm pore membrane, each. The resultant chimeric nanostructures were allowed to anneal for 30 min before measuring them by means of light scattering.

### 3.6. Light Scattering Techniques

The size (hydrodynamic diameter, *D_h_*), size distribution (PDI) and zeta potential (z-pot) of the obtained nanoparticles were investigated through dynamic and electrophoretic light scattering (DLS and ELS, respectively) with a Zetasizer Nano-ZS (Malvern Panalytical Ltd., Malvern, UK) at a detection angle of 90° and were analyzed using the CONTIN method (MALVERN software). For physicochemical properties after preparation, aliquots of the suspensions were diluted 30-fold in PBS medium. Measurements were performed at three different temperature values between 25 °C and 37 °C. In addition, an acidic protocol was performed, by diluting samples 30-fold in citrate buffer 0.1 M (pH = 4.5), allowing them to anneal for 20 min and measuring the size, size distribution and zeta potential at 25 °C. For the evaluation of interactions with serum proteins, static incubation experiments were performed by mixing 100 μL of the prepared chimeric nanosystem dispersions with 100 μL of clarified FBS and incubating them for 30 min at 37 °C. Size, size distribution and zeta potential were measured at 25 °C by diluting the liposome:FBS mixtures 15-fold in PBS medium, to achieve the same biomaterial concentration as that of previous studies. Finally, the physical/colloidal stability of empty chimeric nanocarriers was assessed after 1, 5, 10 and 15 days, by measuring their size and polydispersity.

### 3.7. Transmission Electron Microscopy (TEM)

The morphology of the chimeric nanocarriers was evaluated through negative staining TEM (NS-TEM) analysis. The chimeric liposomes were diluted 60-fold with distilled H_2_O to a final concentration of 333 μg/mL and 5 μL were placed on carbon-coated 200-mesh copper grids (Ted Pella Inc., Redding, CA, USA). The samples were dried at room temperature for 5 min and subsequently stained using uranyl acetate solution for less than 1 min. The samples were examined in a JEM-1010 transmission electron microscope (JEOL Ltd., Tokyo, Japan) at 80 kV. The developed films were scanned using a Perfection V700 PHOTO scanner (Epson, Tokyo, Japan) at a resolution of 1200 dpi.

### 3.8. Drug Entrapment Efficiency % (EE%) and Release Studies

The UV-Vis spectrum of TRAM-34 was obtained with a UV-160A-Vis spectrophotometer (Shimadzu, Kyoto, Japan) and an absorption peak at 261 nm was identified and chosen for further quantification studies. Afterwards, a calibration curve was built in MeOH for drug concentrations between 1 and 250 μg/mL, with correlation analysis leading to R^2^ = 0.9999, utilizing Microsoft Excel (Redmond, WA, USA). The chimeric nanocarriers with incorporated TRAM-34 (0.1 mL samples) were separated from non-incorporated drug molecules by means of size exclusion chromatography (SEC) using a Sephadex-G75 column and utilizing HPLC-grade water as the mobile phase. Empty nanocarriers were used as controls, of which the absorption was subtracted from that of drug-loaded ones, to eliminate false positive results. The isolated liposomes with incorporated drug molecules were diluted with MeOH to 3 mL and measured for absorption at 261 nm. Entrapment efficiency (EE%) was calculated using the following equation:(2)$$IE=\frac{Tram34\left(aftercolumn\right)}{Tram34\left(initial\right)}\times 100\;\;\;\;$$

The release profile of TRAM-34 from EPC:PDMAEMA-b-PLMA 1 or 2 chimeric nanocarriers was studied in four different pH conditions, PBS 0.15 M (pH = 7.4), citrate buffer 0.1 M (pH = 6.5), citrate buffer 0.1 M (pH = 5.5) and citrate buffer 0.1 M (pH = 4.5), at 37 °C. Chimeric nanocarriers with TRAM-34 (0.4 mL samples) were placed in dialysis sacks (molecular weight cut off 12,000; Sigma-Aldrich Chemical Co., St. Louis, MO, USA), which were soaked overnight in the respective medium. Empty nanocarriers were used in all cases as controls. The sacks were then inserted in 8 mL medium, inside a 50-mL falcon, in a Memmert shaking water bath (Memmert GmbH + Co. KG, Schwabach, Germany) set at 37 °C. Aliquots of 0.5 mL were taken from the external solution at specific time intervals and that volume was replaced with fresh medium, in order to maintain sink conditions. The amount of TRAM-34 released at various times, up to 3 h, was determined using the Shimadzu UV-160A-Vis spectrophotometer at 261 nm, with the aid of the calibration curve:(3)$$Tram34\;concentration\left(\frac{ug}{mL}\right)=\frac{absorbance-0.0025}{0.0018}\left(R^{2}=0.9999\right)\;$$

### 3.9. Normal Cell Culture

Human embryonic kidney cells HEK-293 (ATCC, Manassas, VA, USA) were grown in DMEM-Glutamax (Gibco, Thermo Fisher Scientific Inc., Waltham, MA, USA) with 10% heat-inactivated FBS (HyClone, Logan, UT, USA). Cells were routinely maintained on plastic tissue culture flasks and plates (Sarstedt Ltd., Spata, Greece) at 37 °C in a humidified atmosphere containing 5% CO_2_/95% air and subcultured twice a week after 80% confluency was reached.

### 3.10. Normal Cell Viability In Vitro

The toxicity of the chimeric nanocarriers was tested on the HEK-293 human embryonic kidney cell line, utilizing the AlamarBlue^®^/resazurin assay. In the assay, blue non-fluorescent resazurin is reduced to pink, fluorescent resorufin, which is a metabolic response of living cells. This resazurin conversion determines the cell viability [55]. The cells were seeded on a black 96-well plate at a density of 10,000 per well. After 24 h incubation, 20 μL of resazurin solution (1 mg/mL in PBS) was added to each well, and the cells were incubated for 2 h at 37 °C in the dark. Then, resorufin fluorescence was read at λ_ex_ = 530 nm and λ_em_ = 590 nm using a fluorescence microplate reader (Fluoroskan, Thermo Fisher Scientific Inc., Waltham, MA, USA). The cell viability was presented as a percentage of the fluorescence obtained for untreated control cells treated by 1× PBS. Viability was estimated using the following formula:(4)$$Viability=\frac{A}{A_{c}}\times 100\;\;\;\;$$

### 3.11. Confocal Laser Scanning Microscopy (CLSM)

Cell uptake study was performed on the HEK-293 human embryonic kidney cell line, to qualitatively study the internalization of chimeric liposomes dyed with Rhod-PE at an EPC:Rhod-PE molar ratio of 9:0.03. Briefly, cells were plated 24 h before the start of the experiment in 6-well plates containing sterile cover slips (10,000 cells/well). After equilibration, cell uptake of the chimeric liposomes was achieved by adding the samples at a nanocarrier concentration of 30 μg/mL and incubating the cells at 37 °C for 24 h. The cells were then washed three times with PBS (pH 7.4) to eliminate any excess materials and fixed. Finally, the slides were washed twice with PBS and then examined with a TCS SP8 confocal system with LAS 2.0.215022 software (Leica Microsystems, Wetzlar, Germany) equipped with an HC PL APO CS2 63×/OIL objective. The 540-nm supercontinuum white light laser (WLL) (10% intensity) was used as an excitation light source of rhodamine. The emission spectrum was collected at the range of 572–777 nm by means of the conventional detector (PMT). Moreover, transmitted light images were also collected in the same sequence. The confocal scans were performed bidirectionally at a speed of 400 Hz. The fluorescence spectrum was registered from the single confocal plane (pinhole 1.0 AU). Images were collected in logical size format XY 1024 × 1024 pixels, array 0.18 × 0.18 μm. The line average was set at 4 to improve image quality.

### 3.12. Glioma Cell Culture

The GL261 glioma cell line (RRID:CVCL_Y003) was cultured in DMEM, supplemented with 20% heat-inactivated FBS, 100 IU/mL penicillin G, 100 μg/mL streptomycin, 2.5 μg/mL amphotericin B, 2 mM glutamine and 1 mM sodium pyruvate.

### 3.13. Glioma Cell Viability In Vitro

To assess the viability of cells exposed to different concentrations of drug-loaded chimeric nanocarriers, GL261 glioma cells (13 × 10^4^/cm^2^) were treated with empty chimeric nanocarriers EPC:PDMAEMA-b-PLMA 1 or 2 (30 μg/mL) or with TRAM-34-loaded ones (nanocarrier 30 μg/mL, TRAM-34 5 μM) for 24, 48 and 72 h. Cell viability was determined by means of the MTT assay. Results are expressed as percentages of cell survival, with untreated cells presented separately for each timepoint.

### 3.14. Fluorescence Microscopy (FM)

After 24 h of incubation with Rhod-PE-labelled chimeric nanocarriers, GL261 glioma cells were fixed in 4% formaldehyde, washed with Hoechst (1:1000) for 1 h at room temperature for nucleus visualization and analyzed by means of fluorescence microscopy. Images were digitized using a CoolSNAP camera (Photometrics, Tucson, AZ, USA) coupled to an ECLIPSE Ti-S microscope (Nikon, Tokyo, Japan) and processed using MetaMorph 7.6.5.0 image analysis software (Molecular Devices, San Jose, CA, USA).

### 3.15. In Vivo Toxicity

For the in vivo toxicity study, NOD.CB17-*Prkdc^scid^*/J (NOD/SCID) mice, purchased from Jackson Laboratory (The Jackson Laboratory, Bar Harbor, ME, USA), were used. The mouse colony was maintained in a pathogen-free environment in type IIL cages. Male mice, 6–8 weeks old, were used in the studies described here. All animals were kept under specific pathogen-free (SPF) conditions at the animal facility of the Department of Pharmacology, Faculty of Medicine, University of Thessaly (EL42-BIO_Exp03), in a climate-regulated environment (21 ± 1 °C; 50–55% relative humidity), under a 12 h/12 h light/dark circle (lights on at 7:00 AM) and allowed access to food and water ad libitum. Toxicity experiments were performed following the guidelines of the USA National Cancer Institute [56,57,58]. Chimeric liposomes were administered intraperitoneally (i.p.) in the lateral aspect of the lower left quadrant. Acute toxicity studies were carried out for the determination of the single-dose effect of chimeric liposomes. The administered dose was 100, 200, 300, 400 or 500 mg/kg and the effect was observed for 15 days (*n* = 2 mice/dose). For the needs of the experiment the observed and recorded parameters were mortality, body weight loss and behavioral changes.

### 3.16. Statistical Analysis

Results are shown as mean value ± standard deviation (SD) of three independent experiments (*n* = 3). The in vitro toxicity of the chimeric nanosystems on HEK-293 cells comes from three samples (*n* = 3), whereas the in vitro antiproliferative effect of chimeric nanosystems with drug molecules on GL261 glioma cells was the result of four to seven samples (*n* = 4–7). All data were analyzed through one-way ANOVA versus the control and only *p*-values < 0.01 (**) were considered statistically significant.

## 4. Conclusions

The treatment of glioblastoma may be achieved by utilizing biocompatible and functional aDDnSs, such as stimuli-responsive chimeric nanocarriers that will deliver antiglioma agents to the target site and enable their penetration through the glioma cell membrane. In the present investigation, the incorporation of amphiphilic diblock copolymers inside EPC membranes and liposomes was achieved, leading to pH-responsive chimeric nanoparticles, and their evaluation through various techniques was conducted. These included membrane fluidity measurements, physicochemical characterization, pH-responsiveness assessments, the evaluation of interactions with serum proteins and morphology imaging. Through these methods, the thermodynamic, biophysical, physicochemical and morphological properties of these chimeric nanocarriers were delineated. In addition, the in vitro toxicity on normal-like cells demonstrated their biocompatibility and the absence of toxic effects, whereas cellular uptake studies suggested that they penetrate the cell membrane and do not reach the nucleus.

The developed chimeric nanosystems were also utilized as carriers for the antiglioma molecule TRAM-34, presenting a good entrapment efficiency. Combined DSC, DLS and drug release data suggest that the drug release occurs in acidic pH, without alteration of the nanocarrier’s physicochemical properties. The release tests in different pH conditions resulted in different release rates and final released amounts of the drug for physiological and acidic pH values, also depending on the properties of the incorporated copolymer. Although the in vitro efficacy of the drug-loaded nanocarriers was limited, EPC:PDMAEMA-b-PLMA 1 liposomes with TRAM-34 were found to be slightly more effective against GL261 glioma cells and both nanocarriers showed good internalization by the tumor cells after 24 h. Their fine physicochemical properties and colloidal stability, combined with the high endocytosis and potent drug release under acidic conditions, render these pH-responsive nanocarriers interesting as drug delivery systems of TRAM-34 or other drug molecules to glioma cells.

## Figures and Tables

**Figure 1 ijms-22-06271-f001:**
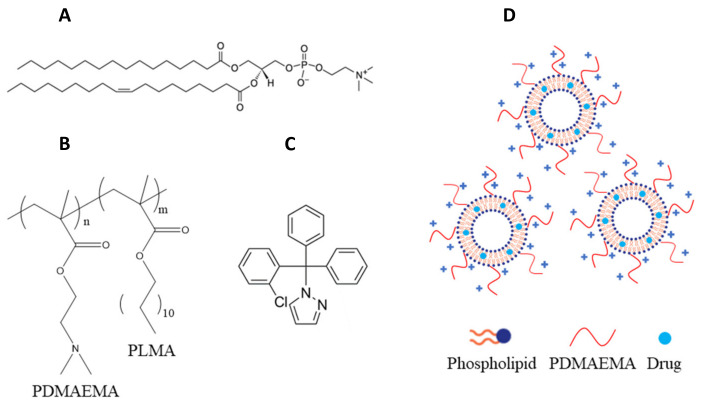
Chemical structures of (**A**) EPC phospholipid, (**B**) PDMAEMA-b-PLMA diblock copolymer, (**C**) TRAM-34 drug molecule and (**D**) the resultant chimeric nanocarriers. The block molar ratio (n:m) in the copolymers is 70–30% for PDMAEMA-b-PLMA 1 and 60–40% for PDMAEMA-b-PLMA 2.

**Figure 2 ijms-22-06271-f002:**
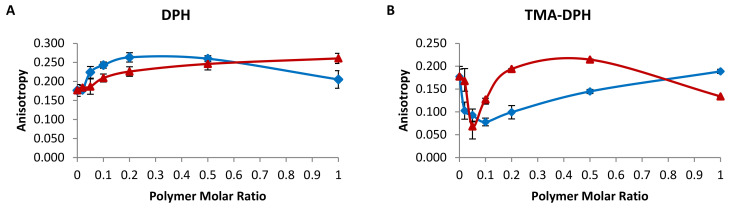
Fluorescence anisotropy of (**A**) DPH and (**B**) TMA-DPH inside EPC membranes, in the presence of increasing molar ratios of PDMAEMA-b-PLMA 1 (blue line) and PDMAEMA-b-PLMA 2 (red line).

**Figure 3 ijms-22-06271-f003:**
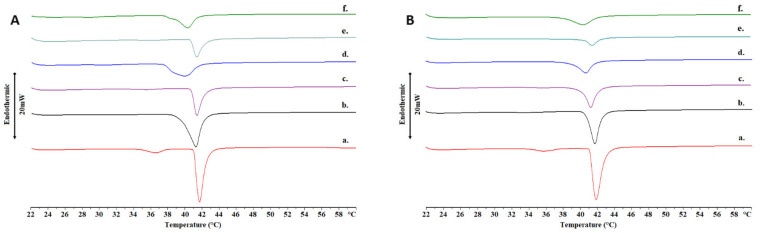
DSC heating curves in (**A**) PBS (pH = 7.4) and (**B**) citrate buffer (pH = 4.5) of a. DPPC, b. DPPC:TRAM-34, c. DPPC:PDMAEMA-b-PLMA 1, d. DPPC:PDMAEMA-b-PLMA 1:TRAM-34, e. DPPC:PDMAEMA-b-PLMA 2 and f. DPPC:PDMAEMA-b-PLMA 2:TRAM-34.

**Figure 4 ijms-22-06271-f004:**
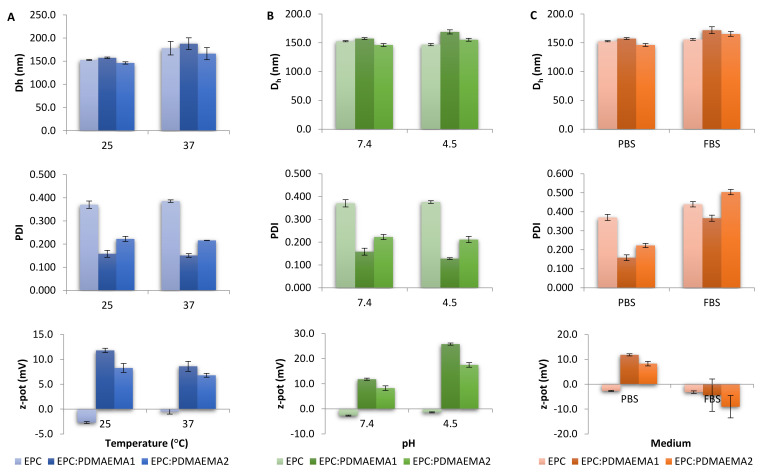
Size (*D_h_*), polydispersity (PDI) and zeta potential (z-pot) of chimeric nanocarriers in different (**A**) temperatures (PBS, pH = 7.4), (**B**) pH conditions (25 °C) and (**C**) media (25 °C).

**Figure 5 ijms-22-06271-f005:**
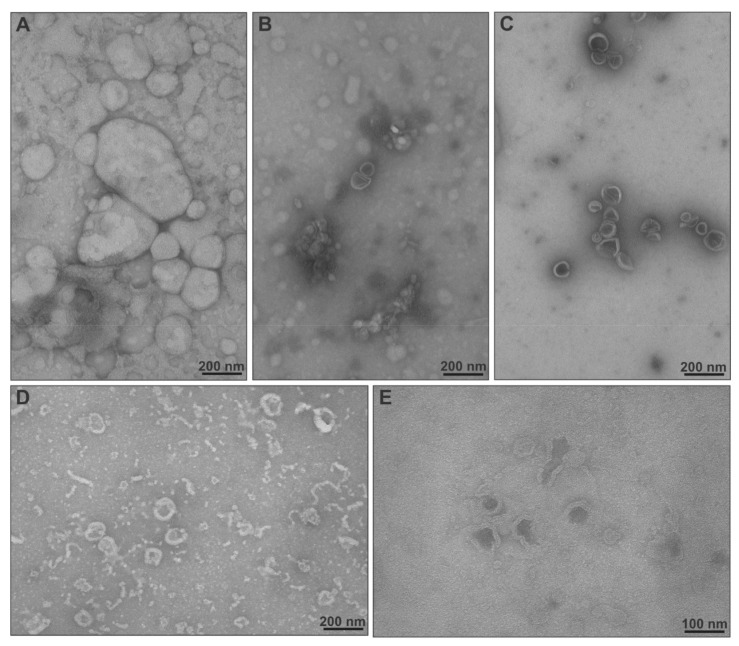
TEM images of (**A**) EPC, (**B**) EPC:PDMAEMA-b-PLMA 1, (**C**) EPC:PDMAEMA-b-PLMA 2, (**D**) EPC:PDMAEMA-b-PLMA 1:TRAM-34 and (**E**) EPC:PDMAEMA-b-PLMA 2:TRAM-34 nanosystems.

**Figure 6 ijms-22-06271-f006:**
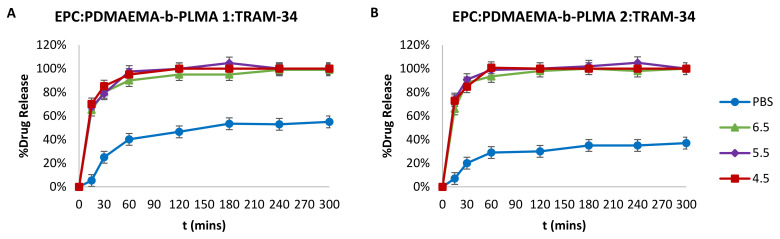
Cumulative drug release of TRAM-34 from (**A**) EPC:PDMAEMA-b-PLMA 1 and (**B**) EPC:PDMAEMA-b-PLMA 2 chimeric liposomes in PBS (pH = 7.4) and citrate buffer (pH = 6.5, 5.5 and 4.5).

**Figure 7 ijms-22-06271-f007:**
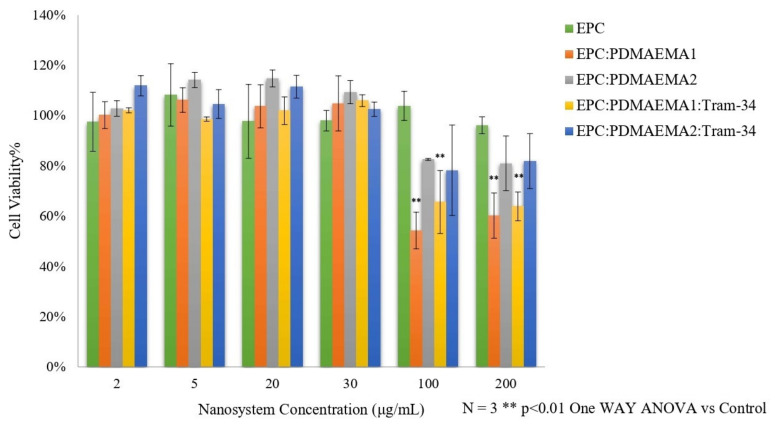
In vitro toxicity of chimeric nanocarriers with or without drug on the HEK-293 cell line.

**Figure 8 ijms-22-06271-f008:**
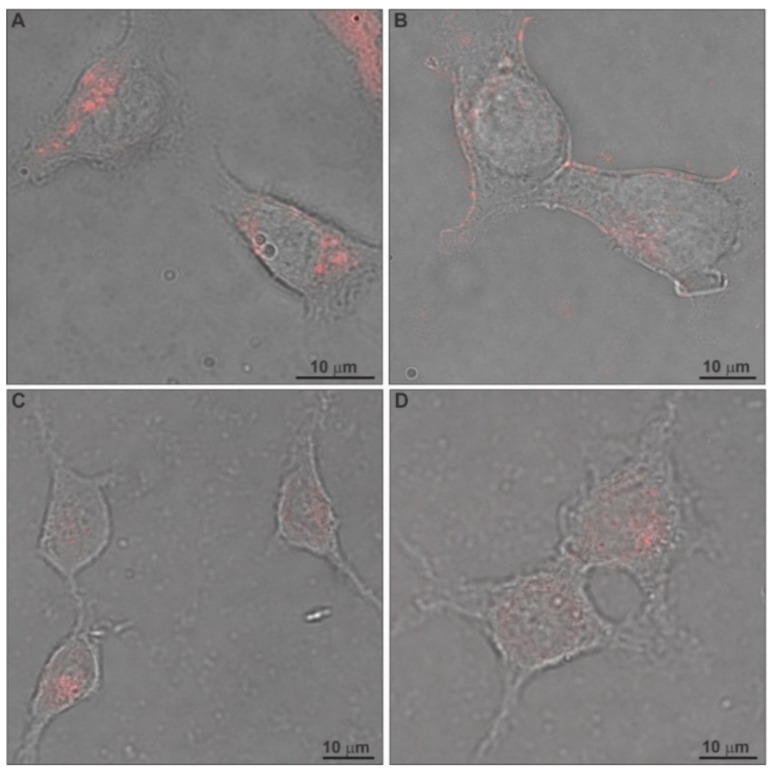
CLSM images of EPC:PDMAEMA-b-PLMA 1 (**A**,**B**) and EPC:PDMAEMA-b-PLMA 2 (**C**,**D**) chimeric nanosystem endocytosis in HEK-293 cells after 24 h incubation at a concentration of 30 μg/mL.

**Figure 9 ijms-22-06271-f009:**
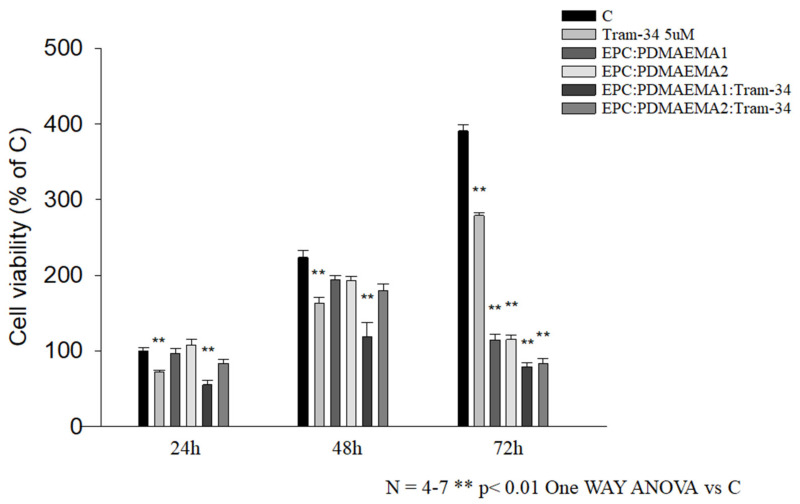
In vitro antiproliferative activity of chimeric nanocarriers with TRAM-34 on the GL261 murine glioma cell line after 24, 48 and 72 h incubation (C stands for control untreated cells).

**Figure 10 ijms-22-06271-f010:**
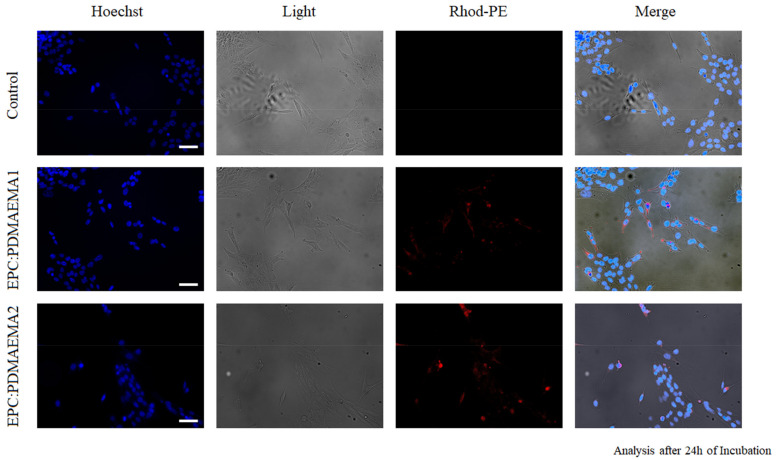
FM images of EPC:PDMAEMA-b-PLMA 1 and EPC:PDMAEMA-b-PLMA 2 chimeric nanosystems in GL261 cells stained with Hoechst (blue) after 24 h incubation at a concentration of 30 μg/mL.

**Table 1 ijms-22-06271-t001:** Physicochemical characteristics of the developed nanosystems.

Nanosystem	D_h_ ^1^ (nm)	SD ^2^	PDI ^3^	SD	z-pot ^4^ (mV)	SD
ΕPC	152.7	1.0	0.370	0.016	−2.7	0.2
ΕPC:PDMAEMA1	157.3	1.7	0.158	0.015	11.8	0.4
ΕPC:PDMAEMA2	146.3	2.5	0.223	0.011	8.3	0.9
ΕPC:TRAM-34	167.7	3.9	0.335	0.022	−1.2	0.1
ΕPC:PDMAEMA1:TRAM-34	164.4	8.0	0.236	0.052	14.0	1.2
ΕPC:PDMAEMA 2:TRAM-34	139.2	0.8	0.193	0.006	12.4	0.8

^1^ Hydrodynamic diameter; ^2^ standard deviation; ^3^ polydispersity index; ^4^ zeta potential.

**Table 2 ijms-22-06271-t002:** IE% of TRAM-34 inside EPC:PDMAEMA-b-PLMA chimeric nanocarriers.

Nanosystem	Initial Concentration (mg/mL)	Concentration before Extrusion (mg/mL)	SD ^1^	IE% ^2^	Concentration after Extrusion (mg/mL)	SD	IE%
EPC:TRAM-34	1.20	0.18	0.03	15	0.00	0.00	0
EPC:PDMAEMA 1: TRAM-34	1.20	0.96	0.06	80	0.88	0.05	73
EPC:PDMAEMA 2: TRAM-34	1.20	0.82	0.05	68	0.74	0.04	62

^1^ Standard deviation; ^2^ entrapment efficiency %.

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
