# Peer review of "Chimeric Stimuli-Responsive Liposomes as Nanocarriers for the Delivery of the Anti-Glioma Agent TRAM-34"

_ijms, 2021, doi:10.3390/ijms22126271_

Round 1
Reviewer 1 Report
The paper is well written and clear. The question asked is of interest, although the methodology only partly answers to the main question of whether this DDS/drug may be interesting in glioma. Generally, this study is a very detailed, well done characterization study of a DDS in vitro, but probably overinterpreted in the efficacy results section. The efficacy results hardly gives insight on its real in vivo potential therapeutic interest in glioma (intracarebral delivey, etc), and superative terms like "great potential" should probably be avoided.
Although the authors claim that they are "notable and always significant" and that is "statistically" true, the results are not sounding in terms of efficacy.
A more "realistic" interpretation/conclusion, i.e. this drug delivey system may be interesting (it is pH responsive and has a good Encapsulation efficacy), it is relatively stable (hours scale), it gets into the cells (internalization inside the cell does not mean that the molecules gets out the endosome, etc) and the efficacy of the model drug in vitro in this system is rather limited.
Overall, these interesting and clear data need to be published but their interpretation and should be more cautious.
Other points :
1) a scheme representing the various DDS tested (and their components) may be of interest to the readers
2) the manuscript may benefit of a some more conciseness and less interpretation in order to be shorter and easier to read.
Author Response
Comment 1:
The paper is well written and clear. The question asked is of interest, although the methodology only partly answers to the main question of whether this DDS/drug may be interesting in glioma. Generally, this study is a very detailed, well done characterization study of a DDS in vitro, but probably overinterpreted in the efficacy results section. The efficacy results hardly gives insight on its real in vivo potential therapeutic interest in glioma (intracarebral delivey, etc), and superative terms like "great potential" should probably be avoided.
Answer 1:
We would like to thank the reviewer for his positive feedback. We agree with his/her suggestion about hyperbolic terms and we revised the manuscript according to that.
Comment 2:
Although the authors claim that they are "notable and always significant" and that is "statistically" true, the results are not sounding in terms of efficacy
Answer 2:
We agree with the reviewer’s comment and a more objective end, such expressions that regard the efficacy have been appropriately replaced.
Comment 3:
A more "realistic" interpretation/conclusion, i.e. this drug delivey system may be interesting (it is pH responsive and has a good Encapsulation efficacy), it is relatively stable (hours scale), it gets into the cells (internalization inside the cell does not mean that the molecules gets out the endosome, etc) and the efficacy of the model drug in vitro in this system is rather limited.
Answer 3:
The results and discussion and conclusion parts have been carefully revised according to the reviewer’s comments.
Comment 4:
Overall, these interesting and clear data need to be published but their interpretation and should be more cautious.
Answer 4:
We have revised the manuscript according to the reviewer’s comment.
Comment 5:
A scheme representing the various DDS tested (and their components) may be of interest to the readers
Answer 5:
The elements suggested by the reviewer are part of the graphical abstract and based on his/her comment, we have added a new Figure 1 with part of that.
Comment 6:
The manuscript may benefit of a some more conciseness and less interpretation in order to be shorter and easier to read.
Answer 6:
We have removed several parts from the results and discussion section and made it more concise according to the reviewer’s comment.
Reviewer 2 Report
This manuscript describes the formulation of pH sensitive liposomes for delivery of TRAM-34 in glioblastoma cells. The nano system is characterized using various techniques, and have shown the proof of concept in mice model.
In general, the study design, result and language are fine. However, the presentation can be improved. The study results are supported by suitable tables and figures, and supplementary material.
Figure 3: labels used in the figure are different from tables. This could lead to confusion. Please make it uniform. Likewise for other figures.
Incorporation efficiency is not a common term. I suggest replacing with entrapment efficiency.
Important control is missing in the in vitro release study. Please include free drug release at same pH.
“Empty nanocarriers were used as reference”, this statement looks incomplete. What does this mean?
What is the relevance of 50oC? Please explain this in manuscript.
Author Response
Comment 1:
In general, the study design, result and language are fine. However, the presentation can be improved. The study results are supported by suitable tables and figures, and supplementary material.
Answer 1:
The manuscript has been revised, especially in the results and discussion and conclusion sections, according to the reviewer’s comment.
Comment 2:
Figure 3: labels used in the figure are different from tables. This could lead to confusion. Please make it uniform. Likewise for other figures.
Answer 2:
We have changed the labels in the tables, so that they are the same with the figures, according to the reviewer’s comment. In addition, we have mentioned the short polymer name the first time the polymers appear in the text.
Comment 3:
Incorporation efficiency is not a common term. I suggest replacing with entrapment efficiency.
Answer 3:
The term has been replace as suggested by the reviewer.
Comment 4:
Important control is missing in the in vitro release study. Please include free drug release at same pH.
Answer 4:
To the best of the authors’ knowledge, the free drug release has not been assessed previously as control. Conversely, the non-loaded nanocarrier is often utilized as such, in order to eliminate any false positive results, in case some nanocarrier components escape the dialysis sack. Some examples are given from De Leo V et al. Molecules. 2018;23(4):739. doi:10.3390/molecules23040739 and Deng W et al. Nat Commun 9, 2713 (2018). doi.org/10.1038/s41467-018-05118-3. In addition, as far as we are concerned, TRAM-34 is a synthetic molecule and has not been formulated inside nanoparticles before.
Comment 5:
“Empty nanocarriers were used as reference”, this statement looks incomplete. What does this mean?
Answer 5:
We have replaced the term “reference” with the more suitable term “control” and added an explanation to this sentence according to the reviewer’s comment.
Comment 6:
What is the relevance of 50oC? Please explain this in manuscript.
Answer 6:
All text associated with 50oC has been removed and Figure 3 has been modified, according the reviewer’s comment.
Round 2
Reviewer 1 Report
The paper was corrected as expected, and is now more concise and clear.
Reviewer 2 Report
Manuscript has been improved.